**Data Availability Statement:** All relevant data are within the paper and its Supporting information files.

# Cross-sectional study of the ambulance transport between healthcare facilities with medical support via telemedicine: Easy, effective, and safe tool

**Carlos H. S. Pedrotti©\*, Tarso A. D. Accorsi, Karine De Amicis Lima, Jose R. de O. Silva Filho, Renata A. Morbeck, Eduardo Cordioli**

Telemedicine Department, Hospital Israelita Albert Einstein, Sao Paulo, Brazil

\* carlos.pedrotti@einstein.br

## Abstract

### Background

Feasibility and safety of ambulance transport between healthcare facilities with medical support exclusively via telemedicine are unknown.

### Methods

This was a retrospective study with a single telemedicine center reference for satellite emergency departments of the same hospital. The study population was all critically ill patients admitted to one of the peripheral units from November 2016 to May 2020 and who needed to be transferred to the main building. Telemedicine-assisted transportation was performed by an emergency specialist. The inclusion criteria included patients above the age of 15 and initial stabilization performed at the emergency department. Unstable, intubated, ST-elevation myocardial infarction and acute stroke patients were excluded. There was a double-check of safety conditions by the nurse and the remote doctor before the ambulance departure. The primary endpoint was the number of telemedicine-guided interventions during transport.

### Results

2840 patients were enrolled. The population was predominantly male (53.2%) with a median age of 60 years. Sepsis was the most prevalent diagnosis in 28% of patients, followed by acute coronary syndromes (8.5%), arrhythmia (6.7%), venous thromboembolism (6.1%), stroke (6.1%), acute abdomen (3.6%), respiratory distress (3.3%), and heart failure (2.5%). Only 22 (0.8%) patients required telemedicine-assisted support during transport. Administration of oxygen therapy and analgesics were the most common recommendations made by telemedicine emergency physicians. There were no communication problems in the telemedicine-assisted group.

**Funding:** The author(s) received no specific funding for this work.

**Competing interests:** The authors have declared that no competing interests exist.

**Abbreviations:** ED, emergency department; IV, intravenous; TM, telemedicine.

## Conclusions

Telemedicine-assisted ambulance transportation between healthcare facilities of stabilized critically ill patients may be an option instead of an onboard physician. The frequency of clinical support requests by telemedicine is minimal, and most evaluations are of low complexity and easily and safely performed by trained nurses.

## 1. Background

Health-system capillarization is associated with greater efficiency in care, mainly by facilitating access to face-to-face care [1]. However, the organization of decentralized emergency services is expensive and complex [2]. Life-threatening cases initially stabilized at a satellite community's emergency department (ED) must be transported to a hospital for complete treatment [3]. Transportation of critically ill patients between hospitals usually requires a highly specialized ambulance team, including a trained driver, paramedics or nurses, and a medical doctor [4]. However, keeping a physician available to assist in such transportation is expensive, expanding on idleness and occupational hazard [5]. Brazil's legislation states that advanced support ambulances must provide medical support [6].

Telemedicine (TM) is an easy, universal, and low-cost tool to solve health-related problems [7]. Evidence suggests that teleconsultation assessment benefits virtually all medical scenarios, including ambulance transportation [8]. In a prehospital setting, the support TM lends to the ambulance team reduces ED referrals [9]. Video communication between the ambulance and the ED may boost the local staff's perception of clinical status and jump-start triage [10]. Remote patient interviews and the interpretation of the conducted tests positively alter disposition patterns [11]. Wireless media communication between the ambulance and remote specialists can be made possible during ongoing transportation [12]. Despite this evidence, no studies have been conducted on the feasibility and safety of ambulance patient transportation with only a remote physician.

Thus, this study aimed to retrospectively analyze the feasibility and outcomes of the ambulance transfer of patients who have been stabilized in a satellite ED to the main hospital with ongoing medical support only via TM. We hypothesized that TM-assisted ambulance transport is feasible and safe, providing background for future controlled studies.

## 2. Methods

This work was approved by the Research Ethics Committee of Hospital Israelita Albert Einstein—reference number 34955620.0.0000.0071. The need for consent was waived by the ethics committee.

It was a retrospective and descriptive study with a single TM center (Hospital Israelita Albert Einstein, São Paulo—Brazil) reference for four satellite EDs of the same hospital. The study population was all critically ill patients admitted to one of the peripheral units from November 2016 to May 2020 and who needed to be transferred to the main building.

TM-assisted transportation was performed using a standard 4G-network-enabled iPad Air $4^{th}$ generation 2020 with a 24x17 cm display and a free version of a HIPAA compliant video-conferencing software from VSee Lab, Inc (Sunnyvale, CA) (Fig 1).

The platform allowed instant internet-based video calls between the ambulance team and telemedicine staff. The iPad was mounted on plastic support attached to the inside roof grab

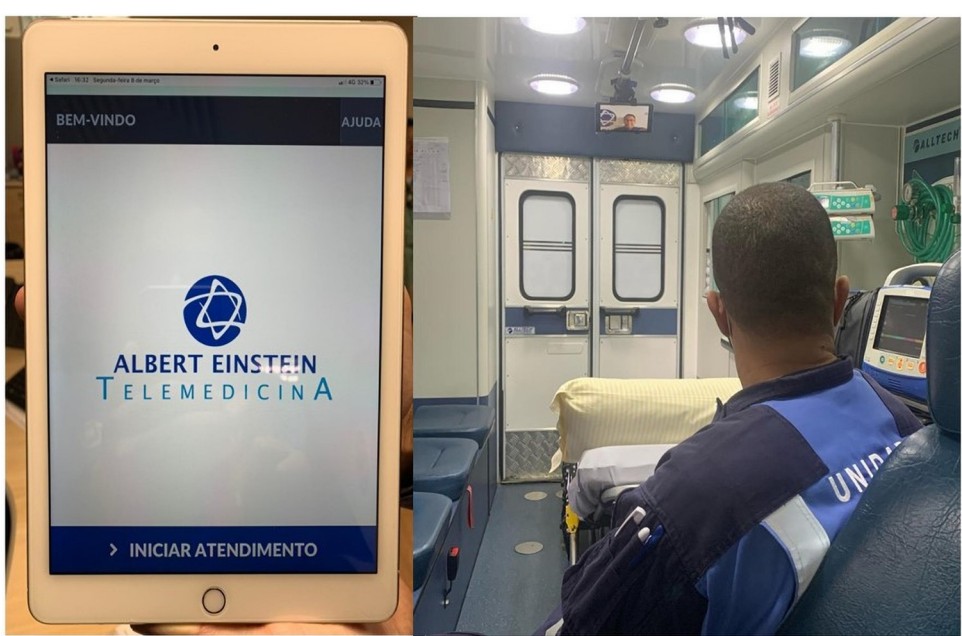

**Fig 1. iPad display with de-identified patient data showed in the used software.**

handles, allowing a comprehensive view of the patient, monitors, and ambulance staff seats. Audio communication was performed using a single-ear Bluetooth headset (Jabra—GN Group–Copenhagen, Denmark) (Fig 2).

The remote assistance was only requested after completing a checklist made by the ambulance nurse confirming patients' stability: no need for advanced ventilation support or vasoactive drugs, no ongoing clinical deterioration, no malignant arrhythmias (Fig 3).

Before departure, a mandatory standardized video call with TM staff was always performed, and the TM physician rechecked the safety protocol and case details, approving or disapproving the departure. TM emergency physicians were available 24/7, and in case of any clinical deterioration after departure, they could be reached immediately by direct video call or by phone if technical problems prevented the video call from happening successfully.

All TM providers on duty at the ambulance assistance service are senior emergency medicine physicians and fully certified in Advanced Cardiology Life Support and other necessary emergency skills to provide emergency healthcare in this institution (accredited by Joint Commission International).

Ambulance routes were standardized, and 4G-signal availability was checked before departure. The distance between the satellites and the central unit varied from 7 to 25 km, with the transport time varying from 10 to 30 minutes. Transport data were electronically recorded in the medical record. Teenagers and adults above the age of 15, to whom hospital admittance to intensive or semi-intensive care, were eligible to TM-assisted transportation. Unstable, intubated, ST-segment elevation myocardial infarction and acute stroke patients were not evaluated since a physician onboard during transportation was required in these situations. The primary endpoint was the number of TM-guided interventions during transport.

After departure, a video call as requested if any monitor alarm was triggered or in case of any clinical deterioration, new patient symptoms, or by ambulance staff clinical judgment. Once continuous TM assistance was requested, the nurse firstly clearly reported the clinical situation to the remote doctor, who was already aware of the patient's previous condition,

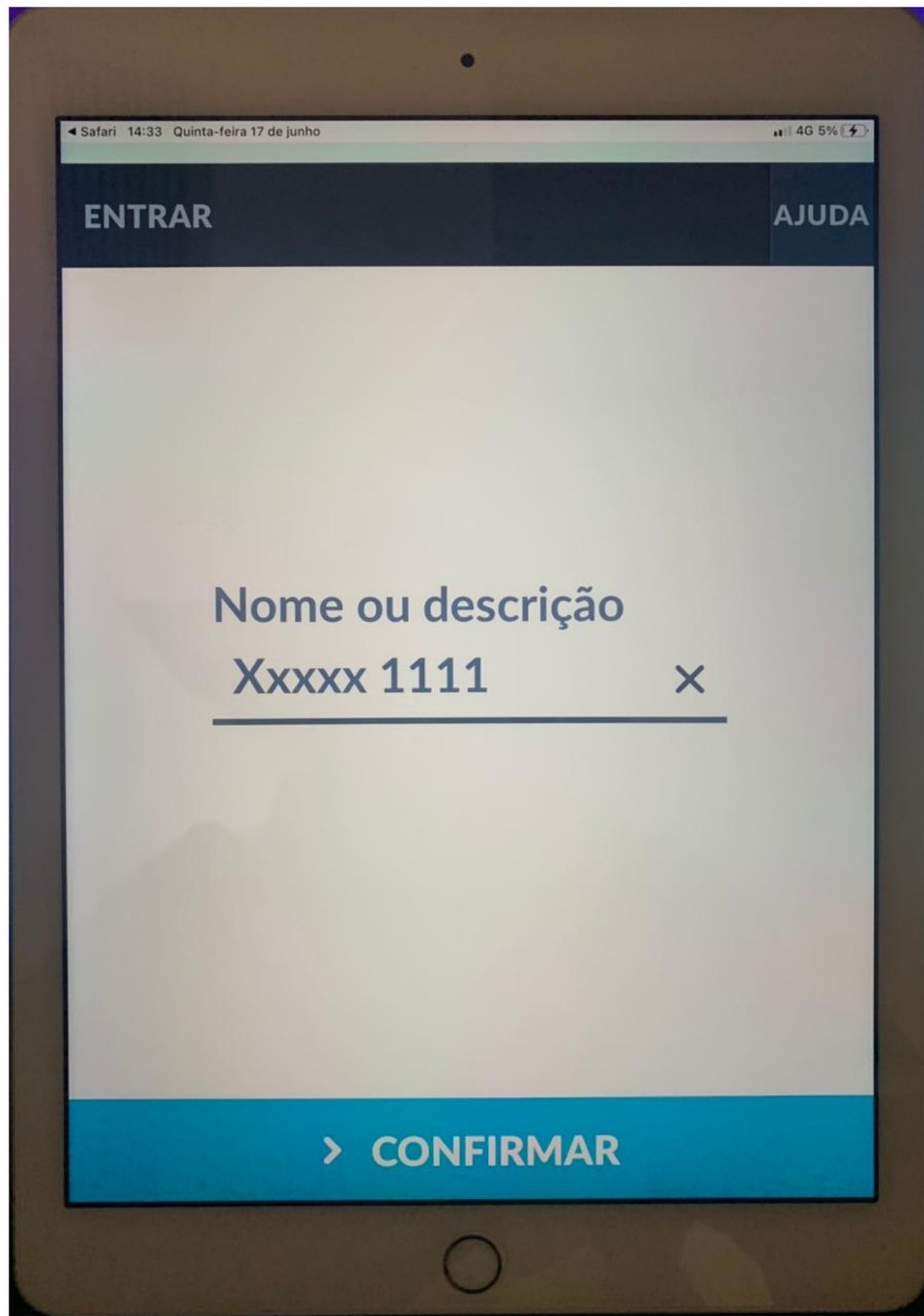

**Fig 2. iPad mounted on plastic support attached to the inside roof grab handles, allowing a comprehensive view of the patient, monitors, and staff.**

followed by a step-by-step approach characterized by: patient-directed camera, vital signs review, and clarification of the current clinical status including direct-to-patient communication if possible. The initial stabilization procedures are guided according to the doctor's judgment, emphasizing maintaining adequate oxygenation and perfusion, followed by symptom control.

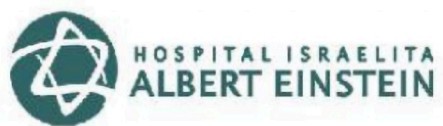

<table>
<tr><td></td><td style="background-color:#f9c9b5">Patient tag</td></tr>
</table>

**RISK ASSESSMENT CRITERIA FOR PATIENT TRANSFER WITH MEDICAL FOLLOW-UP –
ADULT PATIENT**

( ) **Alphaville**   ( ) **Ibirapuera**   ( ) **Perdizes**

Date ____/____/____   Time: Assessment ____:____h Start ____:____h Arrival ____:____h

**Criteria for transferring a patient with mandatory medical follow-up:
This evaluation must be carried out before the transfer.**

| | **Clinical Diagnoses (Adult):** |
|---|---|
| ( ) | Institutional protocols AVC, AMI, Septic Shock and Trauma Code. |
| ( ) | IOT |
| ( ) | Sedated patient or with vasoactive drugs |
| | **Objective Parameters / Instability (Adult):** |
| ( ) | HR ≤ 50 (except athletes) or ≥ 150 bpm |
| ( ) | PAS ≤ 90 / PAM ≤ 60 mmHg |
| ( ) | FR ≥ 40 ipm or Sat O$^2$ < 90% with oxygen support ≥ 50% |

**Vital signs:**

PA ___ x ___ mmHg /   **FC** ___ bpm /   **FR** ___ mrm /   **SatO2:** ___ %

Diagnostic: _______________________________________________

Weight: _____________________________          Bed:________________

( ) Ambulance Advanced - Morumbi Unit (Doctor and Nurse)
( ) Ambulance Morumbi - Intermediate Unit (Nurse and Telemedicine support)
( ) Ambulance Morumbi - Simple Unit

- **All apartment admissions will be by simple ambulance.**
- **Semi, ICU and UPA patients who do not have any of these criticality criteria,
  should come with an intermediate ambulance.**
- **Accommodation is not a criterion for choosing an ambulance.**

| Responsible Dr. | | CRM |
|---|---|---|

Version 4/ Nov / 2016

**Fig 3. Checklist completed before transportation.**

The diagnosis and characterization of the interventions were compiled. The Shapiro-Wilk test was performed for normality analysis. Patient age was described as the median and quartiles and other categorical variables as absolute numbers and percentages. The statistics were only descriptive, and no comparisons were made.

## 3. Results

From 2976 eligible patients, 2840 transportations were analyzed (75 were excluded for STEMI, 53 for acute stroke, 6 for mechanical ventilation or persistent instability, and 2 for the absence of data) from November 2016 to May 2020.

The population was predominantly male (53.2%) with a median age of 60 years, with 25% aged 42 years and 75% aged 67 years.

Sepsis was the most prevalent among the enrolled patients (28%), followed by acute coronary syndromes in patients presenting without persistent ST-segment elevation (8.5%), arrhythmia (6.7%), venous thromboembolism (6.1%), stroke not eligible for thrombolysis (6.1%), acute abdomen (3.6%), multiple causes of respiratory distress (3.3%), and heart failure (2.5%). A small portion of the study population was suffering from poisoning (2%), airway disease (1.9%), and convulsion (0.9). Complete diagnostic data, including less common conditions, can be seen in Table 1.

**Table 1. Demographic data and diagnosis.**

| Variable | Description |
|---|---|
|  | (N = 2840) |
| Age (years), median (Q25, Q75) | 60 (42, 67) |
| Gender |  |
| Female, n (%) | 1329 (46.8) |
| Male, n (%) | 1511 (53.2) |
| Diagnosis |  |
| Sepsis, n (%) | 795 (28) |
| Non-ST-elevation ACS, n (%) | 241 (8.5) |
| Arrhythmia, n (%) | 190 (6.7) |
| Venous thromboembolism, n (%) | 173 (6.1) |
| Stroke not eligible for thrombolysis, n (%) | 172 (6.1) |
| Acute abdomen, n (%) | 102 (3.6) |
| Respiratory distress, n (%) | 94 (3.3) |
| Heart failure, n (%) | 71 (2.5) |
| Poisoning, n (%) | 57 (2) |
| Airway disease, n (%) | 54 (1.9) |
| Convulsion, n (%) | 26 (0.9) |
| Trauma, n (%) | 25 (0.9) |
| Orthopedic, n (%) | 13 (0.5) |
| GO conditions, n (%) | 12 (0.4) |
| Vertigo, n (%) | 11 (0.4) |
| Anaphylaxis, n (%) | 11 (0.4) |
| Metabolic conditions, n (%) | 10 (0.4) |
| Other, n (%) | 773 (27.2) |

ACS, acute coronary syndrome; GO, gynecological and obstetrical.

Only 22 (0.8%) patients required TM-assisted support during transportation. Administration of oxygen therapy and analgesics were the most common TM interventions in 3 (13.6%) patients, intravenous (IV) hydration by hypotension in 2 (9.1%), and antihypertensive treatment in 2 (9.1%). In the other 12 cases, TM was triggered for monitoring evaluation, and there was no need for specific intervention.

Among the TM-assisted patients, 8 (36.3%) were admitted immediately to intensive unit care to complete resuscitation tasks, and among the non-assisted patients, only 5 (0.2%). Just 9 (0.3%) transports experienced partial communication between the ambulance staff and TM (detected in standard communication of arrival at the hospital). There were no communication problems in the TM-assisted group (Table 2).

## 4. Discussion

Interhospital transportation is a joint event in a decentralized health network, and the ambulance transport of critical patients is associated with increased complications [13]. Safety transport needs primary non-human conditions such as appropriate equipment and vehicle and direct handover. However, the ambulance staff plays the most critical role in patient support through intensive monitoring, early red flag recognition, and on-time stabilization [14].

Despite the conceptual importance of experienced staff, novice ED personnel usually are used in ground transport and other hospital doctors upon request [15]. The high cost of maintaining fixed high-quality professionals for transportation, mainly with idle periods and short trips, justify this organization [16]. These new professionals, usually tired and with interruptions in their activity and commute to help with transport, probably have field practice limitations. The temporary absence of these professionals during duty is also associated with crowding in the ED [17]. In parallel, there is evidence to suggest that paramedics (which can be extrapolated to nursing in our country) have discernment in recognizing stable and critically ill patients and can abort transport if there is a high risk of instability. Presumably, these professionals screen for safer transportation situations, implying a low chance of medical intervention on the way [18, 19].

**Table 2. Telemedicine-assisted transport data.**

| Variable | Description |
|---|---|
|  | (N = 2840) |
| TM-assisted support, n (%) | 22 (0.8) |
| TM-guided interventions |  |
| Monitoring evaluation, n (%) | 12 (54.5) |
| Oxygen, n (%) | 3 (13.6) |
| Analgesic, n (%) | 3 (13.6) |
| IV saline solution, n (%) | 2 (9.1) |
| Antihypertensive, n (%) | 2 (9.1) |
| Clinical deterioration after arriving |  |
| TM-assisted support, n/n (%) | 8/22 (36.3) |
| Non-TM-assisted support, n/n (%) | 5/2818 (0.2) |
| Communication difficulties |  |
| TM-assisted support, n/n (%) | 0/22 (0) |
| Non-TM-assisted support, n/n (%) | 9/2818 (0.2) |

IV, intravenous.

Some studies show that up to 15% of patients transported in ambulances arrive at another hospital with hypotension or hypoxia, and part of them have already experienced these changes at the beginning of the transportation and have not been diagnosed. In part, the reason can be attributed to the medical team's characteristics that perform this type of transport, mainly being done by doctors with little experience [20].

It is noteworthy that the occupational risk is also greater in ambulance transport, especially when there is a need to provide care with vehicle movement [21]. Furthermore, approximately 40% of transport is unnecessary, suggesting the need for strict protocols before the patient can board the ambulance [22, 23]. Such evidence presumably supports the low probability of medical intervention during transport. More experienced physicians are an effective measure to ensure transport safety both in the initial assessment and in the handling of possible complications on the way [20]. In this study, the TM physician on duty had experience with critically ill patients, was working with low mental stress conditions, and the aid to transport had little impact on the care routine. This data initially supports the cost-effectiveness of the strategy, with a positive impact on the ED teams who keep their doctors in situ. Furthermore, no impairment was found in TM activity.

Although ambulance transportation of critically ill patients is potentially hazardous, in this study, we did not observe any deaths during transportation, and the frequency of requests for TM support was minimal (0.8%).

After TM connection, the doctor received a bulletin from the nurse according to institutional protocol and specific tasks performed to maintain oxygenation, and organic perfusion was necessary only in 6 patients. Of these, three patients had oximetry quickly stabilized after an increase of nasal oxygen catheter outflow. Another three patients presented hypotension, and the doctor guided IV crystalloid fluid bolus administration. There was no persistent hypotension during transport. Despite the well-managed described situations, placing an advanced airway device by a nurse may be challenging, and predicting fluid responsiveness by TM evaluation during ambulance transportation is nearly impossible. Patients with unstable clinical status can deteriorate quickly and must ideally be transported with a physically present emergency physician. A strict checklist by the ambulance nurse is required before leaving to avoid improper ambulance transport of non-stabilized hypoxemic or shocked patients without an onboard doctor. A double-check is done with the remote TM doctor, who can refuse transportation after mandatory contact among teams before departure. Ambulance staff considered the video calls easy to start using the available software, with no critical technical problems reported. Some transmission delays have been documented, and possibly city regions with tunnels and inadequate climatic conditions should be considered potential hindrances. TM doctors also did not report any relevant trouble performing video calls. There were no reported disruptive behaviors between the ambulance and TM teams.

TM is already a reality in health services, which reduces the time for medical intervention and has a high rate of accuracy in diagnosis and cost-effectiveness [24]. The broad public, professional caregivers, and patients reported a positive attitude toward TM for emergency treatment during ambulance transportation and chronic care at home. These results support further improvement of TM solutions in these domains [25]. Adjusting for health status, socioeconomic status, and provider availability reduced the quartile 1 versus quartile 4 difference in ambulance transport rates. Geographic variability in ambulance use is large and associated with the variation in patient health status and their socioeconomic status [16].

In this study, ambulance transport was considered low risk, with low rates of complications, contrary to what is shown in the literature, and little need for medical intervention. There are two critical points in this finding: 1) the checklist made by the nursing staff who removed extremely unstable patients from this transport modality and 2) the nursing's ability to

recognize a threatening situation, activate TM, and follow recommendations. It is noteworthy that in the study population, only 4.5% of the patients were eligible for the criteria to abort transfer without an onboard doctor. The extremely low rate of patients who became unstable on arrival at the central unit reinforces the effectiveness of the two key points. The vast majority of communications between the ambulance and remote physicians were effective, and there was no compromise in understanding any important recommendation. Only 22 patients needed TM-support intervention during transport, 12 being the only interpretation of data obtained in monitoring. The others had increased oxygen supply, administration of symptomatic drugs, IV hydration, or use of hypotensive agents. About one-third of the patients were unstable when they arrived at the central unit, a situation recognized by changes in vital signs. There was no clear association of these changes with the lack of adequate support in transport; on the contrary, the patients maintained changes similar to the exit from the satellite unit: 3 cases of sepsis with borderline blood pressure, 2 cases of sustained hypertension, and 3 cases of non-critical hypoxemia. This emphasizes the very low number of patients in this situation and without a clear implication of the worst prognosis associated with transport. Although TM is already widespread, there are a few reports in the literature demonstrating the functioning of the interhospital transport system in partnership with TM.

This study demonstrated that telehealth offers a technology strategy to address the potentially unnecessary ambulance transports. Based on prior cost-effectiveness analyses, reducing unnecessary ambulance transports translates into an overall reduction in Emergency Medical System agency costs [26]. Telehealth programs offer a viable solution to support alternate destinations and alternate transport programs [27].

The use of telehealth in transport allows qualified doctors to provide support to several ambulances, reducing the costs for the healthcare system and optimizing team time management; however, it should be noted that there is a need for a well-prepared team (e.g., a qualified nurse). The interventions can be guided, when necessary, via TM without prejudice to the patient. There is already evidence of patients with low clinical severity and non-emergent conditions, and telehealth avoids inappropriate referrals in more than half of the cases. Presumably, these patients will not experience any complications during transportation [27].

Interhospital ambulance transportation is very common nowadays, and despite the assumed association of the need for hospitalization with greater severity, the vast majority of the patients stabilized in satellite units were transported uneventfully to the central unit. A TM-experienced doctor properly guided the very few cases that required any intervention during transport. A checklist before transportation can exclude the cases that need a doctor on board, such as patients with ST-elevation myocardial infarction, ongoing stroke, and intubates.

This study did not accurately assess the cost involved in ambulance transportation without a doctor, and there is no comparison with a standard strategy with the displacement of an ED emergency physician for transport. Assumed cost savings with this transport strategy can be estimated by saving human resources on the institutional payroll. Our institution ED performs about 280 thousand medical encounters per year. Each ED doctor gives care to an average of 1.6 patients per hour, including the night shift. On this average, there is no accounting for the numerous reevaluations of the same patients and bureaucratic activities and contact with other doctors and multi-professional staff. Assuming the absence of TM support, the 2840 transportations during the 43 months analyzed in this study would correspond to at least 113,600 hours of absence from the in situ medical ED (estimated, at best, 40 minutes for the professional to return). The impact is around 88 medical hours consumed per day in transport, clearly implying the need to increase the medical staff.

Meanwhile, the TM center performed around 1,000 encounters a day, and an average of 2 encounters for each doctor per hour, with a less bureaucratic and interprofessional burden. TM time spent on receiving the case report and the arrival notice corresponds to a few minutes, and the need for continuous monitoring of the transport is almost nil. Therefore, ambulance transport with only a TM doctor support sharply saves in situ ED doctors and does not significantly affect the daily TM routine, implying cost containment.

There are some limitations to this study. First, it is a retrospective cohort based on institutional care routine; second, some life-threatening situations have not been contemplated for, and finally, no comparison was drawn with similar groups transported with onboard doctors. Concerning the strength of this study, it reflects real-life practice with 2840 patients with prevalent conditions who were transported safely and with better utilization of physician's time.

## 5. Conclusion

Telemedicine-assisted ambulance transportation from satellite emergency units may be an option instead of an onboard physician in stabilized critically ill patients. The frequency of requested telemedicine support is minimal, and most situations are of low complexity and quickly and safely performed by trained nurses. A rigid stability checklist before departure and standardized communication with the telemedicine center before and after transport are essential for safety. Future controlled studies are needed to address specific clinical conditions, equipment, and safety protocols.

## Supporting information

**S1 Dataset.**
(XLSX)

## Acknowledgments

The authors would like to thank the Telemedicine center IT team and the Emergency Transportation Unit of the Hospital Israelita Albert Einstein for the information technology support and the relevant services provided.

## Author Contributions

**Conceptualization:** Carlos H. S. Pedrotti, Tarso A. D. Accorsi, Karine De Amicis Lima, Eduardo Cordioli.

**Data curation:** Carlos H. S. Pedrotti, Renata A. Morbeck.

**Formal analysis:** Carlos H. S. Pedrotti, Tarso A. D. Accorsi.

**Investigation:** Carlos H. S. Pedrotti, Jose R. de O. Silva Filho, Renata A. Morbeck.

**Methodology:** Carlos H. S. Pedrotti.

**Project administration:** Carlos H. S. Pedrotti, Karine De Amicis Lima, Renata A. Morbeck.

**Resources:** Jose R. de O. Silva Filho.

**Software:** Carlos H. S. Pedrotti, Jose R. de O. Silva Filho.

**Supervision:** Karine De Amicis Lima, Renata A. Morbeck, Eduardo Cordioli.

**Validation:** Jose R. de O. Silva Filho.

**Visualization:** Renata A. Morbeck.

**Writing – original draft:** Carlos H. S. Pedrotti, Tarso A. D. Accorsi, Karine De Amicis Lima, Eduardo Cordioli.

**Writing – review & editing:** Carlos H. S. Pedrotti, Tarso A. D. Accorsi, Karine De Amicis Lima, Eduardo Cordioli.

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
