## [Decision Letter · Decision Letter 0]

15 Feb 2021

PONE-D-20-33545

Cross-sectional study of the ambulance transport between healthcare facilities with medical support via telemedicine: easy, effective, low-cost, and high-security tool

PLOS ONE

Dear Dr. carlos Pedrotti,

Thank you for submitting your manuscript to PLOS ONE. After careful consideration, we feel that it has merit but does not fully meet PLOS ONE’s publication criteria as it currently stands. Therefore, we invite you to submit a revised version of the manuscript that addresses the points raised during the review process.

Interesting article that highlights how telemedicine can be a valid support for facilitating healthcare activities in a hub and spoke logic.

This article retrospectively investigates the role of telemedicine in ambulance transport. Very useful as the scarcity of studies in the field of Telemedicine is known

I believe that it is useful to improve the study by clarifying the following points (by Rewiewer 2 and the Editor):

1. To give more information about the video conferencing software.

2. To improve the discussion about the the specific tasks performed by the EMTs and some critique of problems that they experienced and recommendations for future improvements

3. To specify data about the frequency that TM was required and the types of problems is important and merits a couple of sentences in the abstracts conclusions

4. It is necessary to investigate the issue of cost effectiveness and substantiate the conclusions with data that compare the costs of transport with and without telemedicine.

We look forward to receiving your revised manuscript.

Kind regards,

Filomena Pietrantonio

Academic Editor

PLOS ONE

Additional Editor Comments:

Interesting article that highlights how telemedicine can be a valid support for facilitating healthcare activities in a hub and spoke logic.

This article retrospectively investigates the role of telemedicine in ambulance transport. Very useful as the scarcity of studies in the field of Telemedicine is known

I believe that it is useful to improve the study by clarifying the following points (by Rewiewer 2 and the Editor):

1. To give more information about the video conferencing software.

2. To improve the discussion about the the specific tasks performed by the EMTs and some critique of problems that they experienced and recommendations for future improvements

3. To specify data about the frequency that TM was required and the types of problems is important and merits a couple of sentences in the abstracts conclusions

4. It is necessary to investigate the issue of cost effectiveness and substantiate the conclusions with data that compare the costs of transport with and without telemedicine.

Journal Requirements:

2. In the Methods section, please describe in additional detail the TM- assisted transportation procedure. For instance, 1) the background of the TM emergency specialist, 2) the conditions which have to be met in order for on-going contact with the TM emergency specialist deemed necessary, 3) initial stabilisation procedure.

Furthermore, this is a retrospective study with no control group. As such, we do not feel that any conclusions on the effects of TM-assisted ambulance transportation can be inferred; thus, we ask that you revise the text (especially, but no limited to, the aims and Conclusions) to avoid unsupported statements.

Reviewers' comments:

Reviewer's Responses to Questions

**Comments to the Author**

1. Is the manuscript technically sound, and do the data support the conclusions?

Reviewer #1: Yes

Reviewer #2: Partly

2. Has the statistical analysis been performed appropriately and rigorously? 

Reviewer #1: N/A

Reviewer #2: Yes

3. Have the authors made all data underlying the findings in their manuscript fully available?

Reviewer #1: Yes

Reviewer #2: Yes

4. Is the manuscript presented in an intelligible fashion and written in standard English?

Reviewer #1: Yes

Reviewer #2: Yes

5. Review Comments to the Author

Reviewer #1: This paper provides data on the experience of one health systems with the inter-hospital transport of critically ill patients using telemedicine support (that is, communication with a physician not physically present in the ambulance). The methods and results are clearly communicated and the discussion is appropriate. I have no suggestions on ways to improve the paper.

Reviewer #2: I am really pleased to see this article. We keep hearing a lot about the potential of telemedicine, but there is still relatively little objective data available demonstrating where and how well it works. This seems like an obvious application, which made me wonder why I didn't do it.

I want to say at the outset, I live about a mile from where a medevac helicopter crashed killing a physician and nurse who ere on board. I appreciate you folding this into the discussion.

I found the study methods were straight forward, but I would like a little more information about the video conferencing software. The quality of sound and video can have a significant impact on successful communications. Were the EMTs holding a phone while administering treatment or did they have a headset and video monitor mounted where it could be easily seen? How large was it? Did they have to hold and point a video carmera while administrating treatment?

I think that the results are quite important. Knowledge of what percentage of patients require remote help and the kinds of help that they need are needed to optimize these resources.

I was hoping to see more discussion about the the specific tasks performed by the EMTs and some critique of problems that they experienced and recommendations for future improvements. This would be very helpful to others considering implementation of TM in this setting.

I would like to see a more nuanced conclusion -- especially in the abstract: "Telemedicine-assisted ambulance transportation of stabilized critically ill patients can effectively and safely substitute an onboard physician on most transfers between same-institution locations." This doesn't really provide much useful information. I know that this is an abstract, but that is all that some readers will see. The abstract is a little better at the end, but still falls short in my opinion. "5. Conclusion TM-assisted ambulance transportation of stabilized critically ill patients can effectively and safely substitute an onboard physician on most transfers between same-institution locations. A simple 4G-network-enabled tablet held by plastic support and a Bluetooth headset is enough for effective communication considering the urban environment." As I said above, your data about the frequency that TM was required and the types of problems is important and merits a couple of sentences in your conclusions. Your recommendations for how this system can be improved also could be quite valuable. I also would add a sentence in your conclusion about the safety benefits of TM in this setting.

6. PLOS authors have the option to publish the peer review history of their article (what does this mean?). If published, this will include your full peer review and any attached files.

Reviewer #1: No

Reviewer #2: No

---

## [Author Response · Author response to Decision Letter 0]

7 Apr 2021

We carefully addressed all comments and concerns from the editor and both reviewers and hope the manuscript could be reconsidered for publication. We appreciate all the recommendations, and we are responding each one point-by-point, as required:

1. To give more information about the video conferencing software. 

Response: We very much appreciated this comment and we agree that is necessary more information about the video conferencing software. We inserted the information below in the Methods section:

TM-assisted transportation was performed using a standard 4G-network-enabled iPad and a free version of a HIPAA compliant video-conferencing software from VSee Lab, Inc (Sunnyvale, CA). The platform allowed instant internet-based video calls between ambulance team and telemedicine staff. The iPad was mounted on a plastic support attached to the inside roof grab handles, allowing an ample view of the patient, monitors and ambulance staff seats. Audio communication was performed using a single ear Bluetooth headset (Jabra - GN Group – Copenhagen, Denmark).

The remote assistance was only requested after completion of a check list made by the ambulance nurse confirming patients’ stability: no need for advanced ventilation support or vasoactive drugs, no ongoing clinical deterioration, no malignant arrhythmias. Before departure, a mandatory standardized video call with TM staff was always performed, and TM physician rechecked the safety protocol and case details, approving or disapproving the departure. TM emergency physicians were available 24/7, and in case of any clinical deterioration after departure, they could be reached immediately by direct video call or by phone, if technical problems prevented the video call to happen successfully.

Furthermore, we added a figure (Figure 1): iPad mounted on a plastic support attached to the inside roof grab handles, allowing an ample view of the patient, monitors and staff.

2. To improve the discussion about the specific tasks performed by the EMTs and some critique of problems that they experienced and recommendations for future improvements.

Response: We very much appreciated this comment and inserted the information below in the Discussion section:

After TM connection, the doctor received a boletin from the nurse according to institutional protocol and specific tasks performed aiming maintenance of oxygenation and organic perfusion was necessary only in 6 patients. Of these, 3 patients had oximetry quickly stabilized after an increase of nasal oxygen catheter outflow. Other 3 patients presented hypotension and the doctor guided IV crystalloid fluid bolus administration. There was no persistent hypotension during transport. Despite the well managed described situations, placing an advanced airway device by a nurse may be challenging, and predicting fluid responsiveness by TM evaluation during ambulance transportation is nearly impossible.Unstable patients clinical status can deteriorate quickly and must ideally be transported with a physically present emergency physician. To avoid improper ambulance transport of non-stabilized hypoxemic and/or shocked patients without an onboard doctor, a strict checklist by the ambulance nurse is required before leaving. A double check is done with the remote TM doctor who can refuse transportation after mandatory contact among teams prior to departure. Ambulance staff considered the video calls easy to start using the available software, with no critical technical problems reported. Some transmission delays have been documented and possibly city regions with tunnels and inadequate climatic conditions should be considered as potential hindrances. TM doctors also did not report any relevant trouble performing video calls. There were no reported disruptive behaviors between ambulance and TM teams. 

3. To specify data about the frequency that TM was required and the types of problems is important and merits a couple of sentences in the abstracts conclusions.

Response: We added the information below in the abstract conclusions:

Telemedicine-assisted ambulance transportation between healthcare facilities of stabilized critically ill patients may be an option instead of an onboard physician. The frequency of clinical support requests by telemedicine is minimal and most evaluations are of low complexity and easily and safely performed by trained nurses.

4. It is necessary to investigate the issue of cost effectiveness and substantiate the conclusions with data that compare the costs of transport with and without telemedicine.

Response: We agree that is necessary more information about the cost effectiveness and conclusions about comparing the costs of transport with and without telemedicine. So, we inserted the paragraph below in the Discussion section:

This study did not accurately assess the cost involved in ambulance transportation without a doctor and there is no comparison with a standard strategy with the displacement of an ED emergency physician for transport. Assumed cost savings with this transport strategy can be estimated by saving human resources on the institutional payroll. Our institution ED performs about 280 thousand medical encounters per year. Each ED doctor gives care to an average of 1.6 patients per hour, including night shift. In this average, there is no accounting for the numerous reevaluations of the same patients, as well as bureaucratic activities and contact with other doctors and multi professional staff. Assuming the absence of TM support, the 2840 transports during the 43 months analyzed in this study would correspond to at least 113,600 hours of absence from the in situ medical ED (estimated, at best, 40 minutes for the professional to return). The impact is around 88 medical hours consumed per day in transport, clearly implying the need to increase the medical staff. Meanwhile, TM center performed around 1,000 encounters a day, and an average of 2 encounters for each doctor per hour, with less bureaucratic and interprofessional burden. TM time spent on receiving the case report and the arrival notice corresponds to a few minutes and the need for continuous monitoring of the transport is almost nil. Therefore, ambulance transport with only a TM doctor support sharply saves in situ ED doctors and does not significantly affect the daily TM routine, implying cost containment.

---

## [Decision Letter · Decision Letter 1]

26 May 2021

PONE-D-20-33545R1

Cross-sectional study of the ambulance transport between healthcare facilities with medical support via telemedicine: easy, effective, and safe tool

PLOS ONE

Dear Dr. Carlos Pedrotti,

Thank you for submitting your manuscript to PLOS ONE. After careful consideration, we feel that it has merit but does not fully meet PLOS ONE’s publication criteria as it currently stands. Therefore, we invite you to submit a revised version of the manuscript that addresses the points raised during the review process.

ACADEMIC EDITOR:

The work is interesting and contributes to building the literature on such an important issue for the management of telemedicine in the near future. Before it is published, however, I ask that reviewer 2's suggestions be replied to in a subsequent review:

1) Please include the dimensions of the display and show a couple of typical (de-identified) screen shots to illustrate the type of information being shared.

2) You refer to the checklist used by the nurses. This should be made available online. This is an important aspect of your paper and others will find this useful.

3) page 10: "the doctor received a boletin from the nurse ..." do you mean "Bulletin?"

4) p. 11: "In this study, ambulance transport was considered safe ..." Consider replacing "safe" with "low risk." We are never completely safe.

We look forward to receiving your revised manuscript.

Kind regards,

Filomena Pietrantonio

Academic Editor

PLOS ONE

Journal Requirements:

Additional Editor Comments (if provided):

The work is interesting and contributes to building the literature on such an important issue for the management of telemedicine in the near future. Before it is published, however, I ask that reviewer 2's suggestions be replied to in a subsequent review:

1) Please include the dimensions of the display and show a couple of typical (de-identified) screen shots to illustrate the type of information being shared.

2) You refer to the checklist used by the nurses. This should be made available online. This is an important aspect of your paper and others will find this useful.

3) page 10: "the doctor received a boletin from the nurse ..." do you mean "Bulletin?"

4) p. 11: "In this study, ambulance transport was considered safe ..." Consider replacing "safe" with "low risk." We are never completely safe.

Reviewers' comments:

Reviewer's Responses to Questions

**Comments to the Author**

1. If the authors have adequately addressed your comments raised in a previous round of review and you feel that this manuscript is now acceptable for publication, you may indicate that here to bypass the “Comments to the Author” section, enter your conflict of interest statement in the “Confidential to Editor” section, and submit your "Accept" recommendation.

Reviewer #1: All comments have been addressed

Reviewer #2: All comments have been addressed

2. Is the manuscript technically sound, and do the data support the conclusions?

Reviewer #1: Yes

Reviewer #2: Yes

3. Has the statistical analysis been performed appropriately and rigorously? 

Reviewer #1: Yes

Reviewer #2: N/A

4. Have the authors made all data underlying the findings in their manuscript fully available?

Reviewer #1: Yes

Reviewer #2: Yes

5. Is the manuscript presented in an intelligible fashion and written in standard English?

Reviewer #1: Yes

Reviewer #2: Yes

6. Review Comments to the Author

Reviewer #1: (No Response)

Reviewer #2: Telemedicine is an important topic and is going to become increasingly important in the future. One of the real strengths of this paper is the extensive data set it shares. To many papers only describe the concepts and talk about the possibilities.

I do have a couple of suggestions that should not require further review.

1) Please include the dimensions of the display and show a couple of typical (de-identified) screen shots to illustrate the type of information being shared.

2) You refer to the checklist used by the nurses. This should be made available online. This is an important aspect of your paper and others will find this useful.

3) page 10: "the doctor received a boletin from the nurse ..." do you mean "Bulletin?"

4) p. 11: "In this study, ambulance transport was considered safe ..." Consider replacing "safe" with "low risk." We are never completely safe.

7. PLOS authors have the option to publish the peer review history of their article (what does this mean?). If published, this will include your full peer review and any attached files.

Reviewer #1: No

Reviewer #2: No

---

## [Author Response · Author response to Decision Letter 1]

15 Jul 2021

Reviewer’s comments:

1) Please include the dimensions of the display and show a couple of typical (de-identified) screen shots to illustrate the type of information being shared.

Answer: Display dimension and a screen shot of the de-identified data are included.

2) You refer to the checklist used by the nurses. This should be made available online. This is an important aspect of your paper and others will find this useful.

Answer: The checklist was included as figure 3

3) page 10: "the doctor received a boletin from the nurse ..." do you mean "Bulletin?"

Answer: The correction was made.

4) p. 11: "In this study, ambulance transport was considered safe ..." Consider replacing "safe" with "low risk." We are never completely safe.

Answer: Replacement done.

5) Please provide additional details regarding participant consent. In the Methods section, please ensure that you have specified (1) whether consent was informed and (2) what type you obtained (for instance, written or verbal). If your study included minors, state whether you obtained consent from parents or guardians. If the need for consent was waived by the ethics committee, please include this information.

Answer: The need for consent was waived by the ethics committee. We added this information in the Methods section.

---

## [Decision Letter · Decision Letter 2]

13 Sep 2021

Cross-sectional study of the ambulance transport between healthcare facilities with medical support via telemedicine: easy, effective, and safe tool

PONE-D-20-33545R2

Dear Dr.Carlos Pedrotti ,

We’re pleased to inform you that your manuscript has been judged scientifically suitable for publication and will be formally accepted for publication once it meets all outstanding technical requirements.

Kind regards,

Filomena Pietrantonio

Academic Editor

PLOS ONE

Additional Editor Comments

All comments have been addressed.

This research is interesting and will certainly be followed by increasingly significant studies on the use of telemedicine in the management of complex patients.

Reviewers' comments:

Reviewer's Responses to Questions

**Comments to the Author**

1. If the authors have adequately addressed your comments raised in a previous round of review and you feel that this manuscript is now acceptable for publication, you may indicate that here to bypass the “Comments to the Author” section, enter your conflict of interest statement in the “Confidential to Editor” section, and submit your "Accept" recommendation.

Reviewer #1: All comments have been addressed

Reviewer #2: All comments have been addressed

2. Is the manuscript technically sound, and do the data support the conclusions?

Reviewer #1: Yes

Reviewer #2: Yes

3. Has the statistical analysis been performed appropriately and rigorously? 

Reviewer #1: N/A

Reviewer #2: Yes

4. Have the authors made all data underlying the findings in their manuscript fully available?

Reviewer #1: Yes

Reviewer #2: Yes

5. Is the manuscript presented in an intelligible fashion and written in standard English?

Reviewer #1: Yes

Reviewer #2: Yes

6. Review Comments to the Author

Reviewer #1: No additional comments. No additonal information for above comments. Adding text here to meet minmum requirement.

Reviewer #2: See comments from previous review. Only minor comments were required. This work provides much useful information and provides a good example for study of other medical procedures and topics.

7. PLOS authors have the option to publish the peer review history of their article (what does this mean?). If published, this will include your full peer review and any attached files.

Reviewer #1: No

Reviewer #2: No

---

## [Editor Report · Acceptance letter]

22 Sep 2021

PONE-D-20-33545R2 

Cross-sectional study of the ambulance transport between healthcare facilities with medical support via telemedicine: easy, effective, and safe tool 

Dear Dr. Pedrotti:

I'm pleased to inform you that your manuscript has been deemed suitable for publication in PLOS ONE. Congratulations! Your manuscript is now with our production department. 

Kind regards, 

on behalf of

Dr. Filomena Pietrantonio 

Academic Editor

PLOS ONE